# Five Novel Polymorphs of Cardarine/GW501516 and Their Characterization by X-ray Diffraction, Computational Methods, Thermal Analysis and a Pharmaceutical Perspective

**DOI:** 10.3390/pharmaceutics16050623

**Published:** 2024-05-07

**Authors:** Alexandru Turza, Petru Pascuta, Marieta Muresan-Pop, Liviu Mare, Gheorghe Borodi, Violeta Popescu

**Affiliations:** 1National Institute For R&D of Isotopic and Molecular Technologies, 67-103 Donat, 400293 Cluj-Napoca, Romania; alexandru.turza@itim-cj.ro; 2Department of Physics and Chemistry, Technical University of Cluj-Napoca, 400114 Cluj-Napoca, Romania; petru.pascuta@phys.utcluj.ro (P.P.); liviu.mare@campus.utcluj.ro (L.M.); violeta.popescu@chem.utcluj.ro (V.P.); 3Institute of Interdisciplinary Research in Bio-Nano-Sciences, Babes Bolyai University, 400084 Cluj-Napoca, Romania; marieta.muresan@ubbcluj.ro

**Keywords:** GW501516, cardarine, polymorphism, crystal structure

## Abstract

GW501516, also known by the name of cardarine, is a synthetic peroxisome-proliferator-activated receptor delta (PPR-δ) agonist agent developed for applications in the treatment of metabolic disorders and cardiovascular diseases. A broad polymorph screening in various solvents and mixtures was completed in order to explore its capabilities to grow polymorphs. The crystal structures of four polymorphs were elucidated using single-crystal X-ray diffraction, while one structure was solved via a powder X-ray diffraction method. The solid state features (nature of intermolecular interactions) were investigated by computational methods. The polymorphs were further investigated by thermal DSC analysis and X-ray diffraction on powders. From a pharmaceutical perspective, the stability and solubility of the polymorphs were analyzed as well.

## 1. Introduction

GW501516 (cardarine or 2-[2-methyl-4-[[4-methyl-2-[4-(trifluoromethyl) phenyl]-1,3-thiazol-5-yl]methylsulfanyl]phenoxy]acetic acid) is a compound which belongs to the peroxisome-proliferator-activated receptor delta (PPAR-δ) agonists group [1] and was developed with the purpose of metabolic and cardiovascular disease treatment. A few PPAR-δ ligands have been synthesized so far; these include the title compound as well, and this has been shown to boost the body’s ability to use fat as source of fuel instead of glucose [2,3]. PPAR-δ receptor activation plays a role in metabolism and promotes the burning of fatty acids via the mechanism of up-regulation of fatty acid uptake and oxidation [4,5]. The studies carried out indicate that GW501516 is a compound that shows potential for improvement in the treatment of several conditions such as preventing fatty-acid-induced insulin resistance [6] and treatment of subepithelial fibrosis during asthma [7]. Being a compound that increases the body’s ability to metabolize fats, thus increasing the overall cardiovascular output, it has become popular as a performance-enhancement drug [8,9]. Its use in sports and athletics as an agent to increase performance is prohibited [10].

The study was undertaken with the purpose of investigating the ability of GW501516 (see Figure 1) to form new polymorphs and to investigate them in terms of their structural characteristics. For organic compounds, polymorphism represents the ability of a molecule to configure and pack in two or more distinct ways [11,12]. The preparation and growth of new polymorphs of pharmaceutical active ingredients could offer certain practical advantages in terms of improving solubility, dissolution rate, stability, and a change in the melting point [13]. A search in the CSD database [14] and in the literature showed that the crystal structure for GW501516 and for its polymorphs have not been reported yet. The current study was undertaken with consideration given to the importance of this compound, its promising benefits in pharmaceutics, and the fact that it is slightly soluble in water. It was recrystallized in different solvents and characterized by powder and single-crystal X-ray diffraction, and DSC thermal analysis. For the obtained polymorphs, the analysis of the intermolecular interactions in the lattice and the analysis of the Hirshfeld/fingerprint diagrams were also completed. Moreover, their solubility was measured, and the stability was investigated in special conditions of temperature and relative humidity as well.

## 2. Materials and Methods

### 2.1. Crystallization Experiments

Crystalline white powder of GW501516 was received from Wuhan Shu Mai Technology Co. (Wuhan, China) and solvents from Sigma-Aldrich and Merck, Taufkirchen, Germany. To obtain new polymorphs, a screening was undertaken in a variety of solvents and solvent mixtures. GW501516 (10 mg) with a purity of 99% was dissolved in a wide range of solvents (1 mL) after which the solutions were heated to approximately 40 °C and were allowed to cool to 3 °C. The vials were covered with parafilm in which very small holes were made; the solvents were left to evaporate slowly for several weeks.

The complete list of solvents used is the following: tetrahydrofuran, 2,2,2-trifluoroethanol, 1-butanol, diisobutyl ketone, dimethylformamide, ethyl acetate, 1-propanol, chloroform, 2-ethoxyethanol, 1,4-dioxane, methanol, acetone, nitromethane, ethanol, dichloromethane, dimethylacetamide, dimethylformamide, pentane, acetic acid, methyl isobutyl ketone, diisopropyl ether, and isopropyl alcohol. 

The solvent mixtures used for polymorph screening were 1:1 volumetric ratios of 2,2,2-trifluoroethanol with tetrahydrofuran, methanol with tetrahydrofouran, ethyl acetate with dimethylformamide, chloroform with isopropyl alcohol, ethyl acetate with acetone, diisopropyl ether with acetic acid, and pentane with acetone. 

In this way, the polymorphs were obtained as follows:(i)GW-1 (starting polymorph) from acetonitrile;(ii)GW-2 from mixture of pentane and acetone;(iii)GW-3 in isopropyl alcohol;(iv)GW-5 in methyl isobutyl ketone;(v)GW-4 was found in the bulk of the starting sample as single crystals but was not further reproduced.

### 2.2. X-ray Single-Crystal Diffraction and Structure Refinement

Suitable single crystals were coated in inert oil, attached on a fine nylon loop, and mounted on the goniometer of a SuperNova diffractometer (Rigaku, Tokyo, Japan) which was equipped with dual X-ray micro-sources (Mo and Cu), an Eos CCD detector, and a tube operating at 50 kV and 0.8 mA.

Experimental diffraction intensities were collected and corrected for Lorentz, polarization, and absorption effects in the CrysAlis PRO package [15]. The crystal structures of GW-1 and GW-4 were solved with the SHELXS solution program [16] by Direct Methods, while GW-2 and GW-3 were solved with SHELXT [17] using Intrinsic Phasing. They were refined further via the SHELXL [18] refinement package using Least Squares minimization, all being implemented in the Olex2 package [19]. 

H atoms on carbons were located, refined, and treated by standard riding procedure, considering the isotropic displacement parameter U_iso_(H) = 1.2U_eq_(C) for ternary CH groups [C-H = 0.93 Å] and secondary CH_2_ groups [C-H = 0.97 Å], and 1.5U_eq_(C) considered for all methyl CH_3_ groups [C-H = 0.96 Å]. 

### 2.3. X-ray Powder Diffraction

X-ray diffraction patterns were recorded using a Bruker D8 Advance X-ray diffractometer (Bruker, Karlsruhe, Germany) with a Cu tube operating at 40 kV and 40 mA, and CuKα1 (λ = 1.54056 Å) radiation, which was equipped with a Ge (111) monochromator in the incident beam to obtain only CuKα1 radiation. Data acquisition was undertaken with DIFFRAC plus XRD Commander program at a scan rate of 0.02°/s.

### 2.4. Differential Scanning Calorimetry

The measurement of DSC traces was undertaken with a Shimadzu DTG-60H scanning calorimeter (Shimadzu, Kyoto, Japan). The samples were heated using a closed alumina crucible with a perforated lid with a heating rate of 10 °C/min up to 400 °C under dry nitrogen purge (70 mL/min).

### 2.5. Evaluation of Intermolecular Interaction Energies

Intermolecular interaction energies were calculated using CrystalExplorer software (version 17) [20]. The representative pairs of intermolecular interactions with distances between atoms in different molecules shorter than the sum of the van der Waals radii were calculated by summing the following four energy terms: electrostatic energy (E_ele_), polarization (E_pol_), dispersion (E_dis_), and the exchange repulsion term (E_rep_) [20,21].

The energy terms were calculated using the B3LYP/6-31G(d,p) type wave function. The following scale factors were considered according to the B3LYP energy model: k_ele_ = 1.057k_pol_ = 0.740, k_disp_ = 0.871, and k_rep_ = 0.618.

Hirshfeld surfaces and the corresponding 2D plots were generated with CrystalExplorer software [20] in which the d_norm_ function was used.

The lattice energies were computed by CrystalExplorer [20] using the B3LYP/6-31G(d,p) level of theory by summing the contributions of intermolecular energies for a molecular cluster with a radius of 35 Å.

In all computational methods, the normalized bond lengths (calculated by default in CrystalExplorer) were considered as follows: CH, CH_2_, CH_3_ groups (C–H = 1.083 Å) and O–H distances equal to 0.983 Å based on neutron diffraction [22].

### 2.6. Evaluation of Stability and Preliminary Solubility Check

The stability evaluation was undertaken in the Memmert Humidity chamber HCP105, which is provided with the controlled modification of the relative humidity (setting accuracy of 1% rh) and of the temperature (setting accuracy of 0.1 °C). The samples from the climatic chamber were analyzed from time to time by X-ray diffraction on powders to check possible structural changes.

Solubility was measured at room temperature of 21 °C starting from a certain amount of polymorph over which small volumes of water were successively added. The mixture was stirred for several hours until the solutions became transparent, after which even smaller amounts of the compound were added until it lost its transparency, after which the addition of water was repeated and so on until we were sure that a certain amount of material was dissolved in the solution and no more. For greater precision, two such measurements were made, after which the average value was calculated. 

This was also verified by UV-VIS absorption in the following way. The UV-VIS absorption spectra were obtained for the analyzed polymorphs and the wavelength for which the absorption was maximum was determined. Around the concentration values obtained in the first phase, very small amounts of polymorphs and water were added, making sure that the UV absorption was in the saturation zone of the respective polymorph.

The UV-VIS measurements were performed with the Biochrom PVC Biowave Spectrophotometer (Harvard Bioscience, Holliston, MA, USA), with a 5 nm spectral bandwidth and Xenon source.

## 3. Results

### 3.1. Crystal Structures Descriptions

Detailed crystallographic data and other refinement-related parameters regarding the four polymorphs approached by single-crystal X-ray diffraction are shown in Table 1.

The starting sample consists of a mixture of two polymorphs, namely GW-1 (which is found in the majority) and the minor phase (GW-4) which was selected for single-crystal diffraction analysis, but which was not reproduced further for another analysis.

#### 3.1.1. GW-1

The recrystallization process of GW501516 in acetonitrile yielded the formation of the starting polymorph which crystallized in the centrosymmetric monoclinic P2_1_/c space group. Its asymmetric unit contains one molecule which is built in a two-plane shape at the methylsulfanyl group (Figure 2a). Strong O-H···N hydrogen bonds connect the hydroxyl group with the thiazole rings, which are completed by C-H···O interactions. Oxygen atoms from acetate participate in these interactions, assuring the formation of supramolecular arrangements in layers (Figure 2b). The layers are further stacked by C-H···π interactions between the phenyl ring and the acetate group and show a separation distance of 2.78 Å. The carbon of trifluoromethyl moiety-phenyl-thiazole-methyl-carbon of methylenesulfanyl is in approximately the same plane [(C1)-(C2-C3-C4-C5-C6-C7)-(C8-S2-C10-N1)-(C11)-(C12)] having a root mean square deviation (RMSD) of 0.039 Å. The second plane, having an RMSD of 0.016 Å, is constituted by the sulfur of the methylenesulfanyl-phenoxy-methyl group [(S2)-(C13-14-C15- C16-C17-C18-O1)-C19]. The angle between these two planes is 11.99° and the related torsion angle is C9-C12-S1-C13 is 72.3°. In a similar way, we proceeded to calculate the fit of the planes and the analogous torsion angles for the other polymorphs, the results being presented in Table 2.

#### 3.1.2. GW-2

The slow evaporation of GW501516 in a mixture of pentane and acetone yielded the formation of a new polymorph which crystallizes in the monoclinic crystal system (centrosymmetric P2_1_/n space group) with one molecule in the asymmetric unit, which adopts a twisted shape at the methylsulfanyl and acetate group (Figure 3a). The formation of supramolecular assemblies is driven by a multitude of intermolecular interactions (i.e., C-H···O, C-H···F, C-H···π) among which are strong unidirectional O-H···N hydrogen bonds between the acetate group and the thiadiazol ring (Figure 3b).

#### 3.1.3. GW-3

The polymorph grown by recrystallization in isopropyl alcohol was established to crystallize in the C2/c monoclinic space group with one molecule in the asymmetric unit which adopts a roughly two-plane shape geometry connected by a C12-S2 bond at the methylsulfanyl group (Figure 4a). Strong hydroxyl···thiazole O-H···N bonds combined with C-H···O involving oxygen atoms of the acetate group and C-H···π contacts are involved in overall crystal stability (Figure 4b). It should be noted that the major variation between GW-1 and GW-3 is the rotation of the central heterocycle. Otherwise, the remainder of the structures is very similar.

#### 3.1.4. GW-4

The fourth analyzed and reported polymorph of GW501516, found in the starting sample, crystallizes in a non-centrosymmetric space group (namely Pca2_1_ of the orthorhombic crystal system). In this case, the asymmetric unit is found as a dimer comprising two individual molecules (denoted with A and B suffixes). Both molecules depict two plane-like conformations linked by the C12-S2 bond (Figure 5a). Mutual carboxyl···carboxyl O-H···O hydrogen bonds which form a R_2_^2^(8) homosynthon are present in the dimer formation. Crystal cohesion is assured via strong O-H···O hydrogen bonds between the carboxyl groups, C-H···π interactions between phenyl rings and acetate groups, and C-H···O interaction with the carboxyl oxygen playing the role of donors. The supramolecular packing diagram seen along the *b*-axis is illustrated in Figure 5b. Molecule A and molecule B can be considered to be composed of two fragments bounded by the S2-C12 bond. Their superposition shows that the two molecules are generated approximately from each other by mirroring and the fluorine atoms are rotated relative to each other (Figure 5c).

#### 3.1.5. Crystal Structure Solved by Powder X-ray Diffraction for GW-5

The determination of the crystal structure from the powder diffraction data involves the following steps: indexing the X-ray powder diffraction pattern, Pawley refinement for the correct assignment of the space group, obtaining the structural model, and finally refining the structural model via the Rietveld method [23]. The indexing of the powder diffraction pattern, which aims to determine the crystallographic system and the parameters of the unit cell, was undertaken with the X-Cell program [24] implemented in the Reflex module of the Materials Studio software (version 4.4) [25]. It was found that the polymorph crystallizes in the triclinic system with the following lattice parameters: a = 4.9283 Å, b = 21.2441 Å, c = 21.2749 Å, α = 94.41°, β = 96.34°, γ = 92.84°, and V = 2203.46 Å^3^, with a high figure of merit. Initially we tried to solve the crystal structure from a single crystal, but due to the poor quality of the crystals, we could only go as far as determining the lattice parameters, obtaining a solution like the one obtained from powder diffraction. Considering the density, calculated as 1.369 g/cm^3^, this suggests that there are two molecules in the asymmetric unit. 

The determination of the structural model was based on Monte Carlo simulated annealing and parallel tempering using global optimization techniques with Fox software (version 2022.1) [26]. In this method, the degrees of freedom are changed for each of the two molecules in the asymmetric unit: three degrees of freedom for translations, three for rotations, and eleven for torsion angles, each time calculating the simulated powder diffraction pattern and comparing it with the experimental one. At the end, the configuration with the best fit is retained. 

The obtained structural model was refined via the Rietveld method with the Reflex module of the Materials Studio software [25]. Many parameters on which the diffraction pattern depends have been refined. The diffraction profile was approximated with a Pseudo-Voigt function whose parameters were refined, and the diffraction background was approximated with a polynomial of degree 20. The following parameters were also refined: (U, V, W) from the Caglioti equation [27], the shift parameters (zero point, shift#1 and shift#2), the asymmetry parameters from the Berar–Baldinozzi equation, the preferential orientation parameters (a*, b*, c* and R0) from the March–Dollase approximation, and profile parameters in Bragg–Brentano geometry (NA, NB). In the end, an improved solution was obtained which has a good match between the calculated powder diffraction pattern and the experimental one and which is presented in Figure 6. The final crystallographic parameters are presented in Table 3.

The crystal structures of four polymorphs were determined by single-crystal X-ray diffraction. However, in the case of one polymorph (GW-5, recrystallized in methyl isobutyl ketone) no suitable crystals were grown, thus the crystal structure was elucidated via the powder data technique. 

It was found that this polymorph crystallizes centrosymmetrically in the triclinic P-1 space group with two individual molecules in the asymmetric unit (denoted with A and B suffixes, Figure 7a), exhibiting a high degree of flexibility at the methylsulfanyl moiety. The configuration of the GW-5 polymorph can be described by two planes: one plane is formed by the atoms [(C1)-(C2-C3-C4-C5-C6-C7)-(C8-S2-C10-N1)-(C11)-(C12) and the other from [(S2)-(C13-14-C15- C16-C17-C18-O1)-C19] which are twisted towards each other for molecule A and almost perpendicular to molecule B. For B it can be noted that the acetate fragment is bent as well.

Within the asymmetric unit, the molecules are tightly bonded via hydroxyl···thiazole O3B-H3B···N1A interactions. Other noncovalent intermolecular interactions which are involved in crystal stability are C-H···O, C-H···S and C-H···π contacts.

It can be noted from Figure 7b that structural voids are present and displayed along the *a*-axis and are occupying a volume of 192.8 Å^3^ which consists of a fraction of 8.8% of the unit cell. The computation was done with Mercury software (version 4.0) [28] considering regions within the crystal structure that can host a spherical sample with a 1.2 Å probe radius.

A few conclusions can be pointed out based on the analysis of the structures:(i)GW501516 shows a high ability to be packed in various distinct supramolecular arrangements (polymorphs); the ease of crystallizing in new polymorphs is mainly due to the high flexibility of the carbon-sulfur (C12)-(S2) bond at the methylsulfanyl group and the C20-O1 bond between the acetate group and phenoxy moiety.(ii)GW-1, GW-2, and GW-3 are centrosymmetric, crystallized in the monoclinic system and having the following space groups: (GW-1: P2_1_/c, GW-2: P2_1_/n and GW-3: C2/c). GW-4 is non-centrosymmetric orthorhombic and has space group Pca2_1_ and GW-5 has triclinic space group P-1.(iii)GW-4, found in the start probe, has always recrystallized centrosymmetrically upon dissolution and slow evaporation in various solvents. This is due to the lower overall lattice energy compared with the other polymorphs.(iv)The stability and cohesion in the crystal is mainly ensured as follows: strong O-H···N hydrogen bonds between the carboxyl group and thiazole ring for GW-1, GW-2, and GW-3; strong carboxyl···carboxyl O-H···O hydrogen bonds are found in GW-4 while GW-5 contains both O-H···N and O-H···O interactions. Other weaker interactions that are present in all polymorphs are C-H···π, C-H···F, and C-H···O.

### 3.2. X-ray Powder Diffraction Analysis

The comparison between calculated diffraction patterns generated based on CIF files (Sim) and the experimental patterns of samples grown by recrystallization (Exp) is illustrated in Appendix A. For GW-1, GW-2, GW-3, and GW-5 they show a good match in the position of diffraction lines, which suggests a high purity. However, some diffraction lines in the experimental traces are lower in intensity or are missing, which indicates a certain degree of preferred orientation of crystallites.

The starting sample (Appendix A) comprises a mixture of two polymorphs, namely GW-1 (which is found in majority quantity and whose diffraction peaks are indicated by green stars) and the minor phase (GW-4) which was not further reproduced.

### 3.3. DSC Thermal Analysis

DSC traces related to the polymorphs are presented in Figure 8. For GW-1, the sharp endothermic peak seen at 142 °C is aligned with the melting point of the polymorph while the peak at 82 °C is due to the evaporation of residual solvent left after recrystallization. 

The second polymorph, GW-2, is characterized by a melting point which peaks at 137 °C and polymorph GW-3 shows a melting point of 139 °C. Both samples are solvent free.

Figure 8d presents the DSC curve of the starting material (it comprises a mixture of two polymorphs: in the major quantity, the phase GW-1, and the minor, GW-4). A sharp endotherm at 144 °C is related to the majority phase in the sample (GW-1) whose melting point was assigned in Figure 8a at 142 °C while GW-4, which could not be reproduced, shows a melting point at 131 °C.

The melting point of GW-5 occurs in two steps at a temperature of 139 and 145 °C; prior to its melting point, an endotherm can be seen at 115 °C related to the loss of disordered solvent possibly located inside structural voids. Table 4 shows the enthalpies involved in the thermal events present in the polymorphs.

All five polymorphs exhibit similar thermal behavior at high temperatures and present two exothermic peaks, the first of which is between 270 and 290 °C (associated with degradation and oxidation) and the second between 360 and 370 °C (associated with decomposition and oxidation). 

It is also worth noting that phase transitions may occur under heating or cooling of different polymorphic samples. Such transitions have been reported in the study of metacetamol polymorphs [29]. A more precise evaluation of the stability can be achieved by calculation methods based on DFT. For example, the relative stability of pyrazinamide polymorphs was evaluated using computational methods (DFT) and later confirmed experimentally [30].

### 3.4. Pairwise Intermolecular and Lattice Energies Evaluation

A more in-depth perspective with regard to the solid-state packing in crystal is gained by the computation of intermolecular energies of the molecules within asymmetric units and their neighbors located at distances shorter or equal to the sum of the van der Waals radii. The interaction energy is decomposed in four distinct terms in Appendix A: E_ele_-electrostatic, E_pol_-polarization, E_dis_-dispersion, and E_rep_-repulsion.

The electrostatic interaction is the interaction between the nuclear charges and the electronic densities and they depend on the inverse of the distance between the charges. They are of great importance in molecular crystals that form hydrogen bonds or contain charged groups.

The polarization term represents the effect of an electrostatic field produced by the charge distribution of the polarizing molecule on a charge distribution of a polarized molecule (induced charge dipole) or the interaction between molecules that are characterized by a permanent dipole moment. The polarization energy is proportional to the inverse of the distance to the fourth power.

Temporary fluctuations due to the movement of electrons in molecules lead to the appearance of short-lived dipole moments that produce attractive forces called dispersion between otherwise non-polarized molecules. Although they have a dependence on the inverse of the distance to the sixth power, overall their cumulative effect can be significant, especially for large molecules.

The repulsion energy is due to the partial overlap of the wave functions and electron prevention with parallel spin according to the Pauli principle. This is proportional to the inverse of the distance to the twelfth power.

Thus, some conclusions are drawn related to packaging in polymorphs: (i)Polymorphs GW-1, GW-2, GW-3, and GW-5 show hydroxyl···thiazole O-H···N hydrogen bonds which are characterized by high interaction energies mostly dominated by an electrostatic component (E_ele_).(ii)Polymorphs GW-4 and GW-5 are characterized by an unused nitrogen atom of thiazole rings in the formation of O-H···N hydrogen bonds but form strong mutual carboxyl···carboxyl O-H···O hydrogen bonds with very high binding energy (E_tot_ = −148 kJ/mol) which is overwhelmed by the electrostatic energy (E_ele_ = −114 kJ/mol) in GW-4 and E_tot_ = −140 kJ/mol and with E_ele_ = −106 kJ/mol in GW-5.(iii)Rather low polarization energies (E_pol_) suggest that the molecules are little polarized.(iv)In all five crystals, the stacking by C-H···π contacts are involved mostly in stability and cohesion, an aspect that is observed from the dominant values of dispersion energies (E_dis_) where these interactions are involved; it is observed that in the crystals containing C-H···π interactions, the dispersion energies are much higher.

The relative stability of the five polymorphs was evaluated based on the computation of total lattice energies that is summed in four individual contributions (Table 5).

The GW-4 polymorph is characterized by the lowest lattice energy in absolute value (195.7 kJ/mol), which is in agreement with the lowest melting point which occurs at 131 °C. Furthermore, this is the reason why the polymorph GW-4 could not be reproduced by recrystallization, because the compound tends to pack into forms that have lower overall crystal lattice energy (such as GW-1, GW-2, or GW-3).

The GW-1 polymorph (−231.0 kJ/mol) shows a slightly higher stability than GW-2 (−223.5 kJ/mol), a fact that is also reflected in the melting point of 142 °C for GW-1 compared with 137 °C for GW-2.

Polymorph GW-5, whose melting appears in two steps (139 and 145 °C), is close in its melting point to GW-1, GW-2, and GW-3 and is characterized by a lattice energy of 210.1 kJ/mol in absolute value that indicates a slightly lower stability than these, a fact that can probably be attributed to the presence of voids and possible lack of solvent embedded in the structure that would eventually lead to a decrease in lattice energy and increase in stability. Regarding GW-3, it has a melting point of 139 °C, being close to the rest of the polymorphs, but it has the highest absolute value of lattice energy among all the analyzed structures (236.8 kJ/mol), which is slightly outside what would be expected in terms of correlation with its melting point.

Paying more attention to energy breakdown in individual components, it can be seen that overall the dispersion component has the largest weight in cohesion, followed by the electrostatic one which brings a significant contribution, while the polarization term is almost insignificant. The electrostatic term is the lowest in absolute value in GW-4 due to the fact that the heterocyclic thiazole rings are not participating in O-H···N bonds and GW-2 has the highest electrostatic term, probably due to the bent configuration at the methylsulfanyl and acetate groups.

### 3.5. Hirshfeld Diagrams and Fingerprint Plots Analysis 

The analysis of molecular crystals by Hirshfeld surfaces is a useful tool suitable for the visualization of intermolecular interactions and their impact in crystal structure packing and the supramolecular self-arrangements. The surfaces were generated, mapped, and compared (Appendix A using the d_norm_ function [20]. In the case of GW-4 and GW-5 (Appendix A) separate treatment was required for each molecule comprising the asymmetric unit.

Each intermolecular contact with a distance shorter than or equal to the sum of the van der Waals radii is marked by an arrow and listed in Appendix A.

Based on the color code (red, white, and blue) the surfaces can be explained as follows: red patches correspond to distances shorter than the sum of the vdW radii (strong interactions), white corresponds to distances equal or close to the sum of the vdW radii (weak contacts) and blue denotes longer distances. For each 3D Hirshfeld surface, the related 2D fingerprint plot was generated (Appendix A).

A few conclusions can be pointed out from analyzing the surfaces, the fingerprint plots, and the (d_e_ and d_i_) distances:(i)The fingerprint plots of GW-1, GW-2, and GW-3 are symmetric and specific to the crystals with one molecule in asymmetric units; the fingerprint diagrams of GW-4 and GW-5 are asymmetric, suggesting that the molecular environment is different for each individual molecule in the asymmetric unit (Appendix A).(ii)The presence of sharp and protruding H···N/N···H spikes in GW-1, GW-2, GW-3, and GW-5 is attributable to the strong hydroxyl···thiazole N-H···O interactions which play an important role in packing.(iii)The sharp protruding H···O/O···H spikes in GW-4 and molecule A of GW-5 are representative of the existence of strong carbonyl···carbonyl O-H···O bonds and have major role in overall stability.(iv)The above both pair of spikes (H···N/N···H and H···O/O···H) present in all crystals corroborated with the high absolute values of interaction energies (Appendix A) denote the overall importance in cohesion.(v)A breakdown of fingerprint diagrams (Appendix A) shows the greatest percentage comprises H···H contacts, followed by F···H/H···F, O···H/H···O, C···H/H···C, S···H/H···S contacts. This points to the fact that crystals are governed by hydrogen bonds and combinations of van der Waals interactions.(vi)From the fingerprint plots of GW-5 it can be observed that d_i_ and d_e_ pairs extend to distances up to 3.0 Å and do not fit into the usual two-dimensional geometry of the fingerprint plots [31]. This could be explained by the presence of the solvent in the lattice that was not located by X-ray diffraction calculations on powders. This represents a possibly incomplete crystal structure solving of GW-5. However, the two cardarine molecules are well highlighted.

### 3.6. Stability of Polymorphic Samples and Solubility

The stability of a certain polymorphic form can be considered a shortcoming, because possible phase transitions between polymorphs during a long period of storage can sometimes change the physical state of the drug, thus having the effect of changing the shelf life and effectiveness of the compound. Therefore, storage in the climatic room where the polymorphs are exposed to a certain humidity and temperature regime (relative humidity of 75% and temperature of 40 °C) is necessary to investigate the stability.

The samples were kept for up to two months in the climatic chamber and X-ray diffraction patterns were recorded at certain time intervals (24 h, one week, two weeks, one month, and two months). In Appendix A the comparisons between the patterns are presented. No significant changes were recorded over time, suggesting that the samples are stable and do not undergo phase transitions.

Regarding the solubility of polymorphs in water, we also tried to evaluate the possible advantages of some over the others. The preliminary measurements of solubility show low values, being roughly 75 µg/mL for GW-1, 105 µg/mL for GW-2, 120 µg/mL for GW-3, and 91 µg/mL for GW-5.

For polymorphs GW-1 and GW-2, we tried to improve the solubility by means of a freeze-drying method (for the other polymorphs, we did not try because there was not enough sample material for this method). Polymorphs GW-1 and GW-2 were dissolved in a mixture of ethanol and water and the vials were left open for 24 h, during which a part of the solvent evaporated. Following this, they were exposed to a temperature of −55 °C for two days, after which they underwent lyophilization. The solubility of the samples lyophilized in this way was measured and an improvement was found in the solubility.

Thus, the solubility increased from 75 to 187 µg/mL for GW-1 and from 120 to 212 µg/mL for GW-2.

## 4. Conclusions

The crystal structures of five GW501516 polymorphs were elucidated and reported. Four of them were solved by single X-ray diffraction and one via powder diffraction data using the Rietveld method. GW-1, GW-2, and GW-3 are centrosymmetric monoclinic, GW-4 is non-centrosymmetric orthorhombic, and GW-5 is centrosymmetric triclinic. GW501516 has a high ability to form polymorphs, mainly due to the flexibility of the (C12)-(S2) bond within the methylsulfanyl group and to a lesser extent the flexibility of the (O1)-(C20) bond which connects the acetate group with the phenoxy.

The stability of the crystals is ensured mainly by a dispersion component within which C-H···π contacts have the most important role. The second significant role in cohesion is represented by an electrostatic component which includes strong O-H···N hydrogen bonds between the carboxyl and thiazole rings and reciprocal O-H···O bonds between carboxyl···carboxyl groups. These interactions are also highlighted through Hirshfeld surfaces and fingerprint plots.

DSC analysis concludes that GW-1, GW-2, GW-3, and GW-5 have close melting points and GW-4 (which is present in the starting sample) has a slightly lower melting point. Moreover, the evaluation of the relative stability of the polymorphs through the lattice energy computation is in good agreement with the experimental DSC values of melting points with the exception of GW-3 which has a slightly higher value.

The polymorphs have low solubility in water, which can be improved by lyophilization; the exposure of polymorphs in the climatic chamber and their analysis by X-ray powder diffraction showed they are structurally stable and did not suffer phase transitions.

## Figures and Tables

**Figure 1 pharmaceutics-16-00623-f001:**
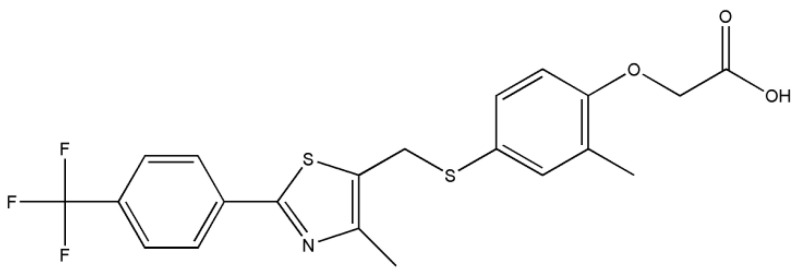
Molecular structure of GW501516.

**Figure 2 pharmaceutics-16-00623-f002:**
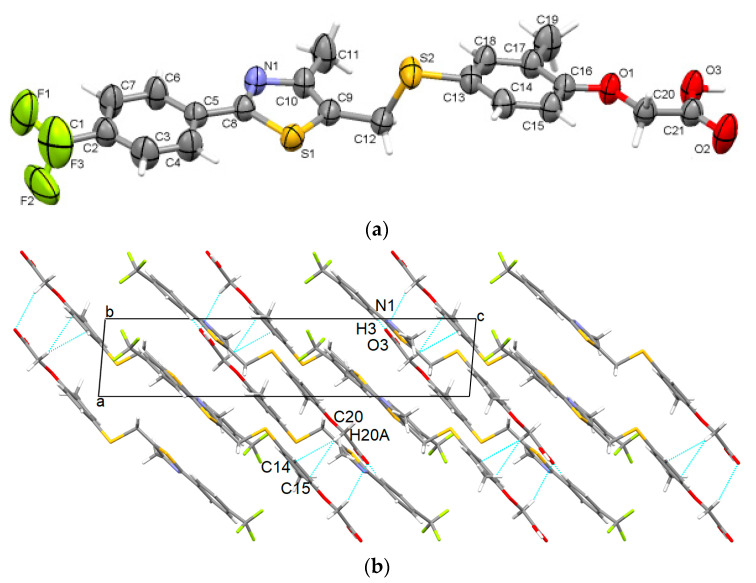
Asymmetric unit of GW-1 illustrating the atoms as thermal ellipsoids at a 50% probability level (**a**); unit cell packing along *b*-axis (**b**).

**Figure 3 pharmaceutics-16-00623-f003:**
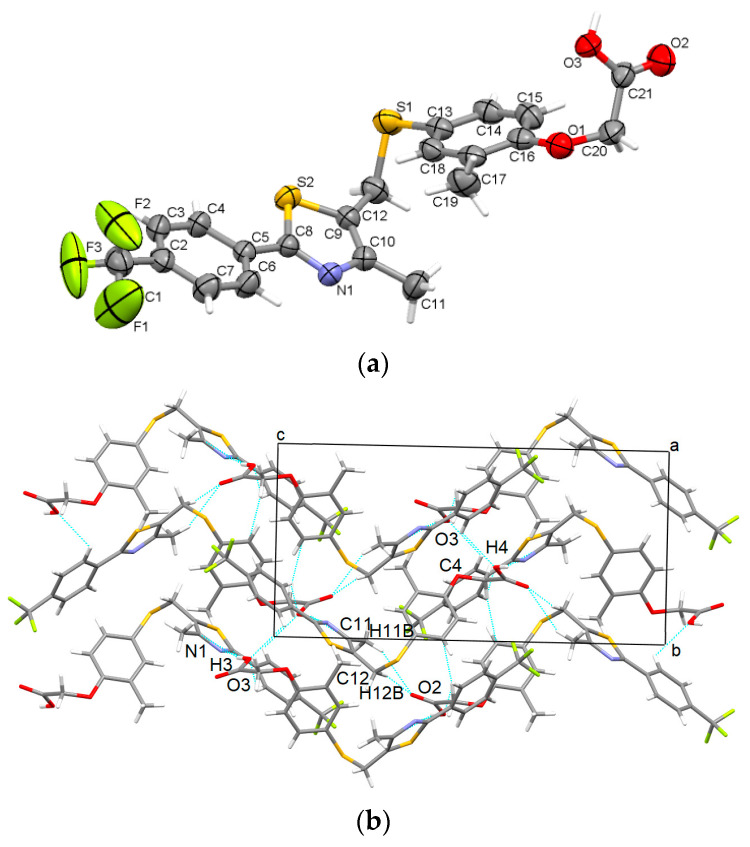
Asymmetric unit of GW-2 illustrating the atoms as thermal ellipsoids at a 50% probability level (**a**); packing perspective along *a-*axis (**b**).

**Figure 4 pharmaceutics-16-00623-f004:**
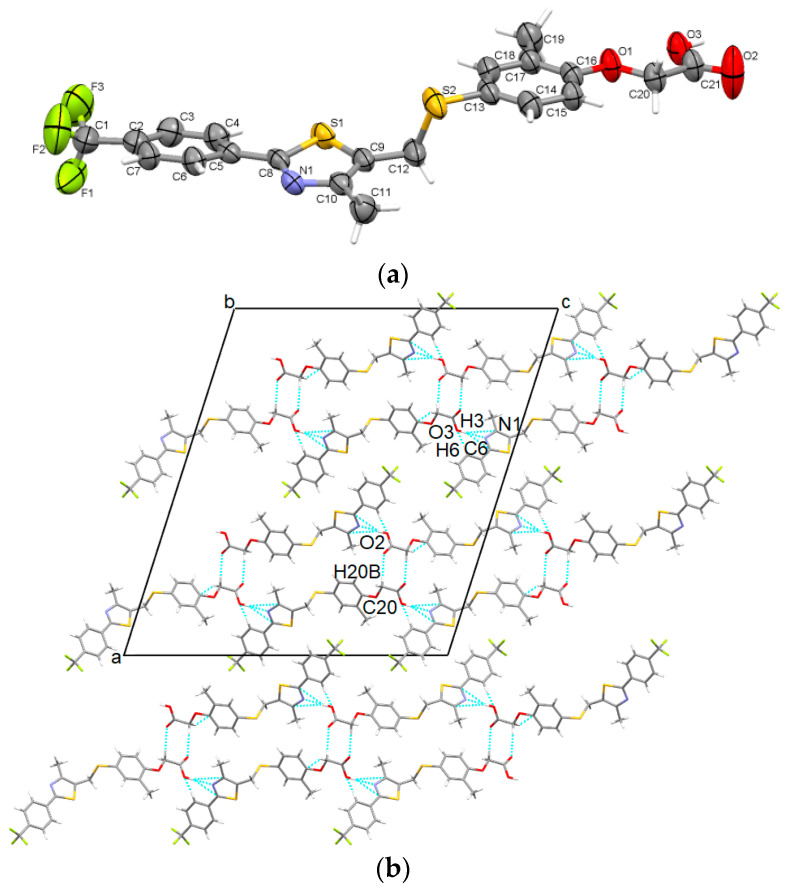
Asymmetric unit of GW-3 illustrating the atoms as thermal ellipsoids at a 50% probability level (**a**); packing perspective along *b-*axis (**b**).

**Figure 5 pharmaceutics-16-00623-f005:**
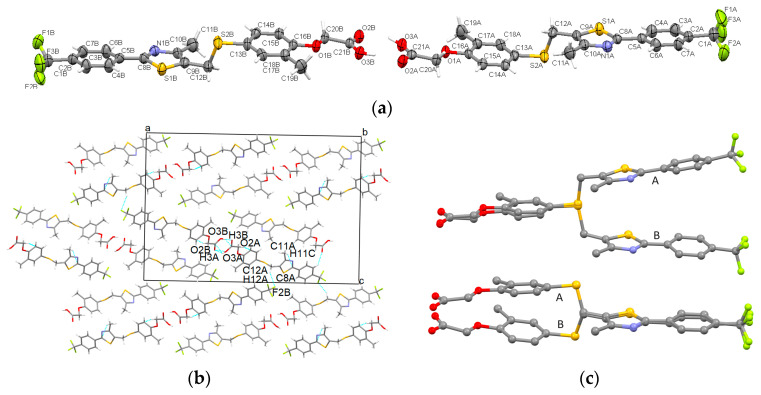
Asymmetric unit of GW-4 illustrating the atoms as thermal ellipsoids at a 50% probability level (**a**); packing perspective along *b*-axis (**b**); superposition of molecules with molecular fragment A and molecular fragment B highlighted (**c**).

**Figure 6 pharmaceutics-16-00623-f006:**
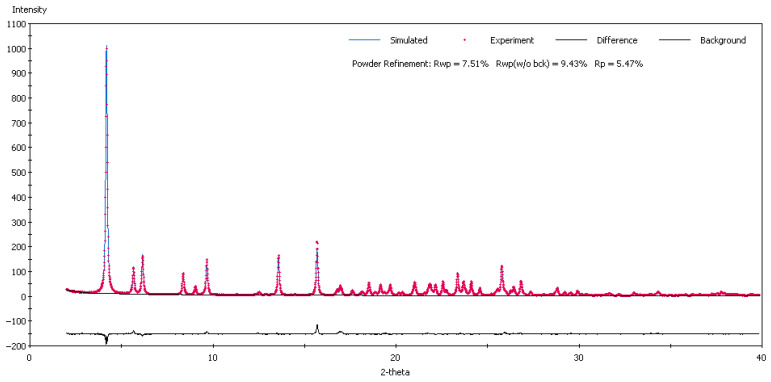
Rietveld refinement match between calculated and experimental X-ray diffraction patterns.

**Figure 7 pharmaceutics-16-00623-f007:**
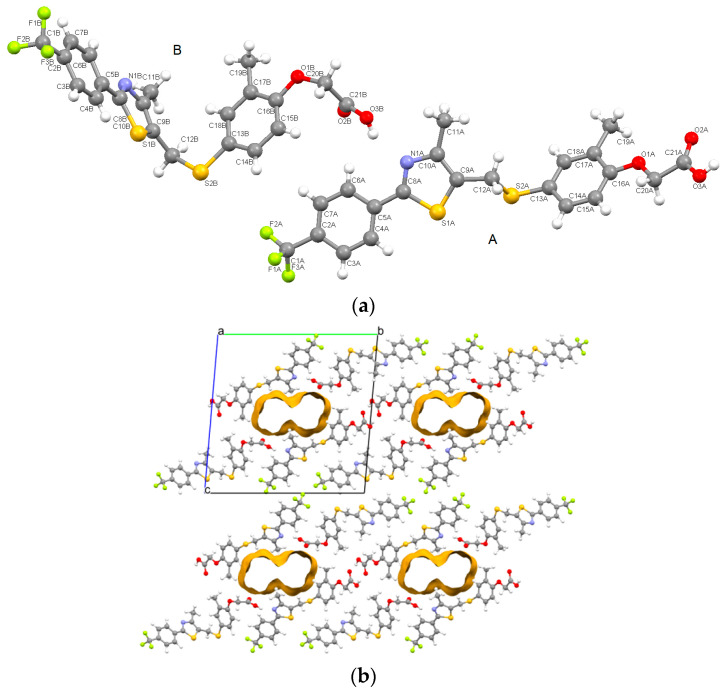
Asymmetric unit of GW-1 displaying the atoms as ball and stick along with the individual A and B molecular fragments (**a**); crystal packing and voids seen along the *a*-axis (**b**).

**Figure 8 pharmaceutics-16-00623-f008:**
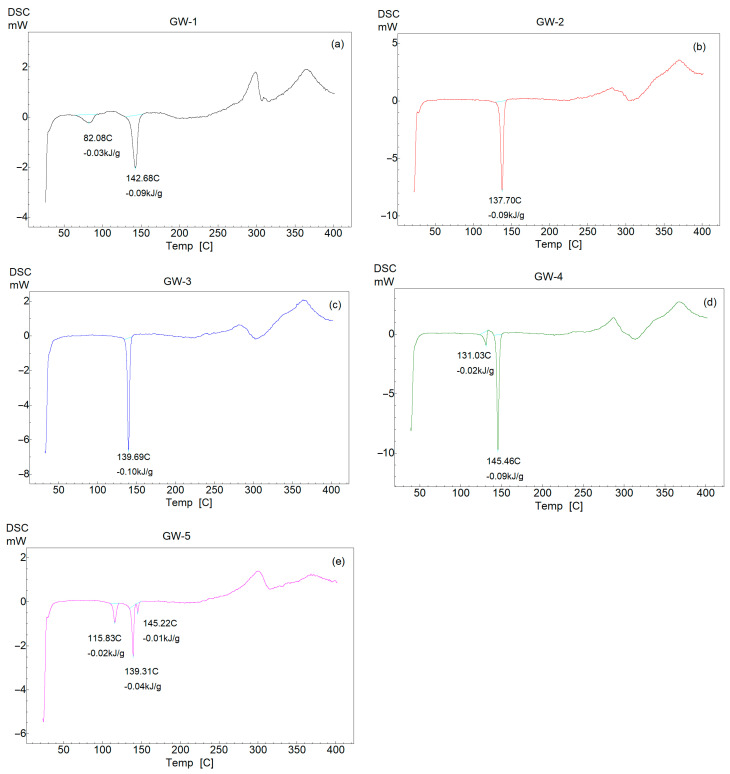
DSC curves of analyzed polymorphs: GW−1 (**a**); GW−2 (**b**); GW−3 (**c**); GW−4 (**d**) and GW−5 (**e**).

**Table 1 pharmaceutics-16-00623-t001:** Detailed single-crystal X-ray diffraction data of polymorphs.

Identification Code	GW-1	GW-2	GW-3	GW-4
Empirical formula	C_21_H_18_F_3_NO_3_S_2_	C_21_H_18_F_3_NO_3_S_2_	C_21_H_18_F_3_NO_3_S_2_	C_21_H_18_F_3_NO_3_S_2_
Formula weight	453.48	453.48	453.48	453.48
Temperature/K	293(2)	293(2)	293(2)	293(2)
Crystal system	monoclinic	monoclinic	monoclinic	orthorhombic
Space group	P2_1_/c	P2_1_/n	C2/c	Pca2_1_
a/Å	4.75852(13)	10.92669(10)	33.1318(18)	35.1542(3)
b/Å	19.4117(6)	9.68349(11)	4.5002(2)	4.906
c/Å	22.6359(6)	19.7719(2)	29.4838(11)	24.1208(3)
α/°	90	90	90	90
β/°	94.909(3)	98.2167(10)	107.691(5)	90
γ/°	90	90	90	90
Volume/Å^3^	2083.23(11)	2070.56(4)	4188.1(4)	4159.78(6)
Z	4	4	8	8
Z’	1	1	1	2
ρcalc g/cm^3^	1.446	1.455	1.438	1.447
μ/mm^−1^	2.769	2.785	2.754	2.773
F(000)	936.0	936.0	1872.0	1872.0
Crystal size/mm^3^	0.2 × 0.1 × 0.1	0.25 × 0.2 × 0.15	0.3 × 0.05 × 0.04	0.25 × 0.15 × 0.10
Radiation	CuKα (λ = 1.54184)	CuKα (λ = 1.54184)	CuKα (λ = 1.54184)	CuKα (λ = 1.54184)
2Θ range/°	6.008 to 141.96	8.758 to 141.298	6.294 to 149.676	6.222 to 141.31
Index ranges	−5 ≤ h ≤ 5, −23 ≤ k ≤ 22, −27 ≤ l ≤ 27	−13 ≤ h ≤ 13, −11 ≤ k ≤ 11, −22 ≤ l ≤ 24	−38 ≤ h ≤ 40, −5 ≤ k ≤ 5, −36 ≤ l ≤ 26	−42 ≤ h ≤ 42, −5 ≤ k ≤ 5, −27 ≤ l ≤ 29
Reflections collected	12,876	29,259	7093	59,180
Independent reflections	3951 [R_int_ = 0.0281, R_sigma_ = 0.0261]	3936 [R_int_ = 0.0247, R_sigma_ = 0.0130]	3938 [R_int_ = 0.0224, R_sigma_ = 0.0365]	7692 [R_int_ = 0.0498, R_sigma_ = 0.0239]
Data/restraints/parameters	3951/0/277	3936/1/274	3938/1/277	7692/3/553
Goodness-of-fit on F2	1.067	1.071	1.188	1.035
Final R indexes [I ≥ 2σ (I)]	R_1_ = 0.0481, wR_2_ = 0.1241	R_1_ = 0.0556, wR_2_ = 0. 1968	R_1_ = 0.0538, wR_2_ = 0.1620	R_1_ = 0.0514, wR_2_ = 0.1377
Final R indexes [all data]	R_1_ = 0.0641, wR_2_ = 0.1360	R_1_ = 0.0625, wR_2_ = 0.2041	R_1_ = 0.0845, wR_2_ = 0.1892	R_1_ = 0.0553, wR_2_ = 0.1435
Largest diff. peak/hole/e Å^−3^	0.35/−0.24	1.04/−0.59	0.54/−0.34	0.98/−0.25
Flack parameter				0.54(2)

**Table 2 pharmaceutics-16-00623-t002:** Specific planes, angles between planes, and torsion angles in polymorphs.

	GW-1	GW-2	GW-3	GW-4	GW-5
Plane P1 and RMSD (Å)	0.039 Å	0.087 Å	0.148 Å	0.046 Å/ 0.052 Å	0.091 Å/ 0.053 Å
Plane P2 and RMSD (Å)	0.016 Å	0.018 Å	0.01 Å	0.07 Å/ 0.011 Å	0.016 Å/ 0.025 Å
Angle between planes P1 and P2	12.0°	40.6°	4.15°	9.2°/ 8.6°	41.6°/ 85.2°
Torsion angle [C9-C12-S2-C13] (°)	−164.5°	72.4°	−163.0°	154.9°/ −151.3°	174.6°/ −75.0°

Abbreviations: Plane P1 [(C1)-(C2-C3-C4-C5-C6-C7)-(C8-S2-C10-N1)-(C11)-(C12)]; Plane P2 [(S2)-(C13-14-C15- C16-C17-C18-O1)-C19].

**Table 3 pharmaceutics-16-00623-t003:** Crystallographic data of GW-5 obtained by X-ray diffraction analysis.

Polymorph	GW-5
Chemical formula	C_21_H_18_F_3_NO_3_S_2_
Formula weight (g/mol)	453.48
Crystal system	triclinic
Space group	P-1
Z	4
Z^’^	2
a (Å)	4.9088(7)
b (Å)	21.269(3)
c (Å)	21.312(4)
A (°)	94.476(13)
Β (°)	96.387(13)
γ (°)	92.795(12)
V (Å^3^)	2200.82
R_wp_ (%)	0.0751
R_p_ (%)	0.0547
ρ_calc_ (g/cm^3^)	1.369

**Table 4 pharmaceutics-16-00623-t004:** Enthalpies involved in thermal events.

Polymorph	Temperature (°C)	Enthalpy (kJ/mol)
GW-1	82	−13.60
	142	−40.81
GW-2	137	−40.81
GW-3	139	−45.34
GW-4	131	−9.06
GW-5	115	−9.06
	139	−18.13
	145	−4.53

**Table 5 pharmaceutics-16-00623-t005:** Crystal lattice energies of polymorphs correlated to melting points.

Crystal	E_ele_ (kJ/mol)	E_pol_ (kJ/mol)	E_disp_ (kJ/mol)	E_rep_ (kJ/mol)	E_latt_ (kJ/mol)	Melting Point (°C)
GW-1	−77.1	−21.9	−189.4	57.4	−231.0	142
GW-2	−114.1	−18.2	−183.5	92.3	−223.5	137
GW-3	−91.9	−19.9	−176.8	51.8	−236.8	139
GW-4	−63.9	−14.6	−171.7	54.5	−195.7	131
GW-5	−84.4	−19.7	−151.6	45.6	−210.1	139 and 145

Abbreviations: E_ele_-electrostatic; E_pol_-polarization; E_disp_-dispersion; E_rep_-repulsion; E_latt_-total lattice energy.

## Data Availability

CIF files of GW501516 polymorphs were deposited via the Cambridge Crystallographic Data Centre: 2331530 (GW-1); 2331529 (GW-2); 2331528 (GW-4); 2331527 (GW-3); 2331526 (GW-5). The copies can be obtained free of charge on written application to CCDC, 12 Union Road, Cambridge, CB2 1EZ, UK (fax: +44 1223 336033); on request via e-mail to lexandru.turza@itim-cj.ro or by access to http://www.ccdc.cam.ac.uk (accessed on 30 April 2024).

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
