# Peer review of "Five Novel Polymorphs of Cardarine/GW501516 and Their Characterization by X-ray Diffraction, Computational Methods, Thermal Analysis and a Pharmaceutical Perspective"

_pharmaceutics, 2024, doi:10.3390/pharmaceutics16050623_

Round 1
Reviewer 1 Report
Comments and Suggestions for Authors
The submitted work described the combined experimental and theoretical analysis of an important API. What is more, the Authors have obtained 4 unique polymorphic form of this compound, which is a great accomplishment. The presentation, however, should be improved. The detailed comments can be found below.
Title of the work is not detailed enough. I suggest adding the acronyms of the methods used to study the polymorphism and the name of the compound – cardarine. Also the fact that 4 forms have been obtained for the first time should be emphasized.
Line 38, this statement needs a reference
Line 55, this is not a chemical perspective but a molecular structure
Line 60, please provide the complete list of those solvents and their mixtures
Lines 75-76, what about GW-5? And also, why two different methods of solving the structure have been used?
Line 90, why the TGA analysis has not been done since it is a complementary method?
Lines 95-107, for the future, please use the periodic DFT, I suggest CASTEP. This is more accurate than the current approach.
Figure 5, the superposition of the conformations should be added to clearly show the differences between them
Line 213, how the density and Z’ are connected to each other?
Table 2, were the predicted (calculated) unit cell dimensions compared with the ones obtained from SCXRD, as described in line 211?
Lines 298-315, please suggest the stability order of the polymorphs
Author Response
Dear Reviewer, Thank you for your overall attention and time to improve the manuscript. All the indicated points have been addressed and we hope that they have led to the improvement of the revised manuscript.
The submitted work described the combined experimental and theoretical analysis of an important API. What is more, the Authors have obtained 4 unique polymorphic form of this compound, which is a great accomplishment. The presentation, however, should be improved. The detailed comments can be found below.
Title of the work is not detailed enough. I suggest adding the acronyms of the methods used to study the polymorphism and the name of the compound – cardarine. Also the fact that 4 forms have been obtained for the first time should be emphasized.
Answer: The title was changed as follows: “Five novel polymorphs of cardarine/GW501516 and their characterization by X-Ray diffraction, computational methods, thermal and a pharmaceutical perspective”
Line 38, this statement needs a reference
Answer: A new reference for this statement was added: DOI10.1002/dta.3013
Line 55, this is not a chemical perspective but a molecular structure
Answer: This was corrected
Line 60, please provide the complete list of those solvents and their mixtures.
Answer: The complete list of solvents was provided.
The complete list of solvents used is the following: tetrahydrofuran, 2,2,2-trifluoroethanol, 1-butanol, diisobutyl ketone, dimethylformamide, ethyl acetate, 1-propanol, chloroform, 2-ethoxyethanol, 1,4-dioxane, methanol, acetone, nitromethane, ethanol, dichloromethane, dimethylacetamide, dimethylformamide, pentane, acetic acid, methyl isobutyl ketone, diisopropyl ether, isopropyl alcohol.
The list of solvent mixtures used for polymorph screening is mixture of 2,2,2-trifluoroethanol with tetrahydrofuran, methanol with tetrahydrofouran, ethyl acetate with dimethylformamide, chloroform with isopropyl alcohol, ethyl acetate with acetone, diisopropyl ether with acetic acid.
Lines 75-76, what about GW-5?
Answer: GW-5 was solved using X-Ray diffraction on powders. This is why it is not included at lines 75-76.
And also, why two different methods of solving the structure have been used?
Answer: We did our best to solve the crystal structure of GW-5 by single crystal X-Ray diffraction but the crystals were not good enough for analysis. This is why we approached polymorph GW-5 and solved its structure by powder X-Ray data.
The following statement is included in the manuscript at section 3.1.5. “Initially we tried to solve the crystal structure from single crystal, but due to the poor quality of crystals, we could only go as far as determining the lattice parameters, obtaining a solution like the one obtained from powder diffraction.”
Line 90, why the TGA analysis has not been done since it is a complementary method?
Answer: We did not have enough material for all polymorphs and a bit more sample is needed for our TGA.
Lines 95-107, for the future, please use the periodic DFT, I suggest CASTEP. This is more accurate than the current approach.
Answer: We will try for future papers to use DFT method and CASTEP program for molecular computation.
Figure 5, the superposition of the conformations should be added to clearly show the differences between them
Answer: The superposition of conformations was added. Molecule A and molecule B can be considered to be composed of two fragments bounded by the S2-C12 bond. Their superposition shows that the two molecules are generated approximately from each other by mirroring and the fluorine atoms are rotated relative to each other.
Line 213, how the density and Z’ are connected to each other?
Answer: Since the density of the other polymorphs is 1.4 g/cm3, then for the triclinic unit cell (space group P-1) four cardarin molecules would be needed in the cell to obtain a density of 1.4 g/cm3. So, the asymmetric unit (Z') would contain two distinct molecules and a final calculated density is 1.37 g/cm3.
Table 2, were the predicted (calculated) unit cell dimensions compared with the ones obtained from SCXRD, as described in line 211?
Answer: Yes, by SCXRD indexing it was obtained a triclinic unit cell with similar lattice parameters as those determined by powder data.
Lines 298-315, please suggest the stability order of the polymorphs
Answer: Based on melting points, the stability of polymorphs in increasing order is GW-4, GW-2, GW-5, GW-3 and GW-1 which is the most stable polymorph but differences are not remarkable.
Reviewer 2 Report
Comments and Suggestions for Authors
Dear authors,
Please find Report attached,
Best wishes

Comments on the Quality of English LanguageDear colleagues,
English needs some minor polishing.
Best wishes,
Author Response
Dear reviewer 2, we have attached a Word document with the answers.

Reviewer 3 Report
Comments and Suggestions for Authors
The authors present their study of polymorphs of the PPRdelta receptor agonist GW501516. The significance of the work is established in the introduction. The experimental work appears to be carried out carefully and thoroughly. I have a few small suggestions for improvement:
1. Figure 2: after the first sentence of the caption, the designator (a) should be included.
2. From the figure of GW-1 in comparison to the other figures, it appears that the dihedral angle C9-C12-S2-C13 is listed incorrectly as 72.3°. That dihedral visually appears to be closer to 170° similar to that of GW-3. It is certainly not a shallower dihedral than GW-2!
3. It should be noted that the major variation between GW-1 and GW-3 is the rotation of the central nitrogen/sulfur heterocycle. Otherwise, the remainder of the structures are very similar.
4. Line 210: it would be helpful to comment more thoroughly as to why the crystals for GW-5 were insufficient for single crystal x-ray analysis like the others. What made them of poor quality and was there no way to correct for it?
5. Figure 6: Is it possible to displace the simulated diffraction pattern above the experimental to allow for a closer examination of the two patterns?
6. Line 307: Please comment on why the data for GW-4 cannot be reproduced.
7. Line 317: should be "intermolecular"
8. Section 3.4: It would be helpful to briefly explain what each of the interaction terms measures for the decomposed energies. This would help with understanding as to why the polarization energies are so low.
With appropriate attention to the above suggestions, I would support publication of this paper.
Comments on the Quality of English Language
Some minor editing of english will be required, but overall the manuscript is very readable.
Author Response
Dear reviewer 3, we have attached a document with the answers.

Round 2
Reviewer 1 Report
Comments and Suggestions for Authors
The Authors have revised and improved their work. This version can be accepted.
Author Response
Dear reviewer #1,
Thank you for your help in the process of reviewing our manuscript.
Reviewer 2 Report
Comments and Suggestions for Authors
Dear authors,
Thanks for your revision. Please, find new comments in the attached file.
Best wishes.

Comments on the Quality of English LanguageEnglish should be revised. Some of the sentences have very specific wording or constructions (e.g. title of the paper whcih is not structured ideally).
Author Response
Dear reviewer #2, we did our best to implement all the valuable suggestions given. We would like to thank you once again for your involvement in the revision of our manuscript.
Please see the file attached.
